# Mitigation of CyanoHABs Using Phoslock^®^ to Reduce Water Column Phosphorus and Nutrient Release from Sediment

**DOI:** 10.3390/ijerph182413360

**Published:** 2021-12-18

**Authors:** Ji Li, Kevin Sellner, Allen Place, Jeffrey Cornwell, Yonghui Gao

**Affiliations:** 1School of Oceanography, Shanghai Jiao Tong University, Shanghai 200030, China; liji81@sjtu.edu.cn; 2Center for Coastal & Watershed Studies, Hood College, Frederick, MD 21701, USA; sellner@hood.edu; 3Institute of Marine and Environmental Technology, University of Maryland Center for Environmental Science, Baltimore, MD 21202, USA; place@umces.edu; 4Horn Point Laboratory, University of Maryland Center for Environmental Science, Cambridge, MD 21613, USA; cornwell@umces.edu

**Keywords:** Phoslock^®^, cyanobacteria bloom, phosphate removal, nutrient fluxes, coupled nitrification–denitrification

## Abstract

Cyanobacterial blooms can be stimulated by excessive phosphorus (P) input, especially when diazotrophs are the dominant species. A series of mesocosm experiments were conducted in a lake dominated by a cyanobacteria bloom to study the effects of Phoslock^®^, a phosphorus adsorbent. The results showed that the addition of Phoslock^®^ lowered the soluble reactive phosphate (SRP) concentrations in water due to efficient adsorption and mitigated the blooms. Once settled on the sediments, Phoslock^®^ serves as a barrier to reduce P diffusion from sediments into the overlying waters. In short-term (1 day) incubation experiments, Phoslock^®^ diminished or reversed SRP effluxes from bottom sediments. At the same time, the upward movement of the oxic–anoxic interface through the sediment column slightly enhanced NH_4_^+^ release and depressed N_2_ release, suggesting the inhibition of nitrification and denitrification. In a long-term (28 days) experiment, Phoslock^®^ hindered the P release, reduced the cyanobacterial abundance, and alleviated the bloom-driven enhancements in the pH and oxygen. These results suggest that, through suppression of internal nutrient effluxes, Phoslock^®^ can be used as an effective control technology to reduce cyanobacteria blooms common to many freshwater systems.

## 1. Introduction

Harmful cyanobacterial blooms (CyanoHABs), which cause negative impacts on ecosystems and human health, are increasing in both frequency and magnitude in aquatic systems [1,2]. This increase is closely associated with eutrophication and climate change [3,4]. Excessive nutrients due to changes in land use and human activities have increased the bioavailability of nitrogen (N) and phosphorus (P). With climate change, rising temperature and stronger stratification may create a more suitable environment for cyanobacteria [5]. CyanoHABs, therefore, are expected to become more frequent and severe in the future [6,7]. The prevention, control, and mitigation of CyanoHABs have drawn increased attention in recent years.

Some cyanobacteria species can fix N to satisfy their demands and release fixed N to support the non-N_2_-fixing species [8,9]. Limited nitrogen availability may favor diazotrophic cyanobacterial species, which would account for high populations in the phytoplankton community [7]. The reduction of point source P inputs through phosphate detergent bans and improved wastewater P removal leads to improved water quality [10,11,12,13]. In addition, such a decrease in P loading and internal P sources via sediment dredging have proven to be viable options to mitigate CyanoHABs in lakes and estuaries [14,15,16]. However, the depuration of stored sediment P is often slow, and that pollution legacy, combined with fresh additions of labile organic and inorganic P, provides a buffer against P depletion. Whereas ecosystems remove N by denitrification, phosphorus removal occurs primarily through burial in sediments [17,18]. In summer conditions, organic matter decomposition and decreased sediment redox conditions result in increased P releases, which support high rates of primary production in the water column, often as cyanobacteria. Low redox conditions, increased algal P deposition, and the elevation of pH by CyanoHAB photosynthesis can promote sediment P release [19,20,21], reinforce eutrophication, and stimulate help to sustain CyanoHAB blooms [22].

Phoslock^®^, a novel lanthanum (La)-modified bentonite, adsorbs soluble phosphate by forming LaPO_4_. The compound is insoluble from pH 4 to 10 [23], with higher adsorption capacity at higher temperatures [24]. Changes in anoxic/oxic conditions exert little effect on the LaPO_4_ stability, minimizing P escape from reducing sediments into the overlying waters [25]. Owing to these properties, Phoslock^®^ can help in the control of algal blooms by reducing the bioavailable P in water [26] and effectively retaining P in sediments [25,27]. Several studies have reported the successful utilization of Phoslock^®^ during algal blooms [25,28], though limited observations of sediment nutrient release are available [29,30,31]. In previous studies, the assessment of Phoslock^®^ application has focused on its performance (direct P adsorption and P removal efficiency in wastewater treatment) [25,27]; the equilibrium and kinetics strength of La-P in response to water column changes (e.g., temperatures, ionic strength, and pHs) [25]; and the reduced P release after capping on the sediment in the field [31]. There is a general consensus that prevention is the preferred management strategy for HABs, but this management strategy is usually difficult to implement. The whole ecosystem responds after the application of Phoslock^®^ remains questionable. In order to provide a sustainable solution for bloom control, we consider both the N and P internal cycles, the induced N:P stoichiometry changes, and the mechanisms regulating phytoplankton biomass and speciation. The long-term impacts on the ecosystem after Phoslock^®^ application are also assessed.

In the present study, Phoslock^®^ was applied in mesocosms to reduce phosphorus concentrations and control cyanobacteria in a freshwater lake subjected to reoccurring cyanobacterial blooms. To assess the performance of Phoslock^®^ as a mitigation strategy, we studied changes of soluble reactive phosphate (SRP) along with chlorophyll *a* (Chl *a*) after applying Phoslock^®^. Further, we estimated how Phoslock^®^ application changed the benthic fluxes of dissolved inorganic nutrients (i.e., soluble reactive phosphorus (SRP), NH_4_^+^, NO_3_^−^, and NO_2_^−^); denitrification; oxygen consumption (influx of O_2_); and the efflux of dissolved inorganic carbon (DIC).

## 2. Materials and Methods

Higgins Mill Pond (38.519 N, −75.964 W), a 152-acre freshwater lake located in Maryland, USA, is located at the headwaters of the Transquaking River, a Chesapeake Bay tributary. The typical water depths are ~1.0–1.5 m. The pond is surrounded by farmland, mostly corn crops and poultry houses, yielding substantial and continuous nutrient loading from fertilizer and poultry litter applications. A confined chicken feeding operation ~0.5 km from the lake and runoff from an upstream chicken processing plant likely led to low N:P of the receiving water, resulting in reoccurring cyanobacterial blooms.

### 2.1. Experiment Design

Two sets of mesocosm experiments were conducted to evaluate the impacts of Phoslock^®^ (SePRO Corporation, Whitakers, NC, USA) on the CyanoHABs and the P in the water and sediment of Higgins Mill Pond (Figure 1). In the first field experiment, Phoslock^®^ was applied in closed-bottom mesocosms to evaluate the impacts on the water column in July when a cyanobacteria bloom dominated the pond. In the second experiment, Phoslock^®^ was sprayed in open-bottom limnocorral as a capping reagent on the sediment. On the 28th day after Phoslock^®^ application, sediment cores were collected from the treated and untreated sites to compare changes in the sediment–water biogeochemical exchange rates. A one-day Phoslock^®^ treatment experiment was also performed to determine the short-term impacts on benthic chemical fluxes.

### 2.2. Phoslock^®^ Impacts on the Water Column

To isolate the nutrient input from the sediment, six closed-bottom mesocosms (30.5 cm inner diameter × 61 cm high) were installed in the lake: 3 for Phoslock^®^ treatment and 3 as untreated controls. On 2 July, 3 mesocosms were filled with 40 L of lake water, and then, each was amended with 1 L of fresh water mixed with 60-g Phoslock^®^. Changes of the Chl *a,* salinity, temperature, pH, and dissolved oxygen (DO) were measured on day 1, day 3, and day 9 after treatment using a Yellow Springs Instrument (YSI) 6600 Data Logger. On day 9, water samples were filtered through Whatman^®^ glass fiber (GF/F) filters and preserved in acid-washed bottles at −20 °C until nutrient analysis. The water samples were also fixed with Lugol’s acid solution to quantify the phytoplankton abundance.

### 2.3. Phoslock^®^ Effects on Sediment Nutrient Processes

To evaluate the performance of Phoslock^®^ on P retention and N cycling in the sediments, 1-kg Phoslock^®^ was spread evenly into an open-bottom 2-m diameter limnocorral (3.14 m^2^, Curry Industry, Ltd., Winnipeg, MB, Canada) on 26 July with a water depth of ~1.5 m. Samples were collected 1 to 2 times per week from ~0.2 m until 12 September for Chl *a*, DO, pH, and dissolved nutrients. Lugol’s acid-preserved phytoplankton samples were processed weekly to identify the cyanobacteria species and estimate the cell abundance.

On 22 August, a pole corer was used to collect 7-cm inner diameter, 15-cm deep acrylic sediment cores for the pore water analysis, with 4 cores collected in the limnocorral and 7 from untreated areas. At the laboratory, the sediment was hydraulically extruded and placed in 10-mL centrifuge tubes under N_2_ gas. Intact sediment cores, taken from treated and untreated sites, respectively, were sectioned at the depth intervals of 0–0.5 cm, 0.5–1.0 cm, 1.0–1.5 cm, 1.5–2.0 cm, 2.0–3.0 cm, 5.0–7.0 cm, and 9.0–11.0 cm.

For the other Phoslock^®^ addition experiment, 3 of the 7 sediment cores from the untreated area had 2.5-g Phoslock^®^ (=0.32 Kg m^−2^) added, with a 4-h settling period for the particle sedimentation. All sediment cores, including 3 after 28 days of Phoslock^®^ exposure, 3 from natural sediments, and 3 with the new Phoslock^®^ addition, were submersed in filtered field water (0.5-µm fibrous polypropylene string-wound cartridges, Cole-Parmer Inc., Vernon Hills, IL, USA) from the bottom of the lake and gently bubbled with air overnight in the dark at 27.5 °C, the ambient temperature at Higgins Mill Pond, in a temperature-controlled incubation chamber. Following the protocol of Gao et al. (2012) [32,33], all the cores were sealed with acrylic O-ring tops with a magnetic stir bar suspended beneath it and a rotating magnetic turntable with a stir bar below the resuspension thresholds. Water-only incubation cores were used as controls to separate water column processes from biological influences, such as the microbial uptake of nutrients, respiration, and nitrogen fixation. During the 6-h incubation, the overlying water was sampled 4 times, with samples filtered through GF/F filters for dissolved inorganic nutrients (SRP, NH_4_^+^, NO_2_^−^, and NO_3_^−^), and a whole water sample preserved with 10-mg HgCl_2_ in a stoppered glass tube for analysis of the dissolved gases (N_2_, O_2_, and DIC) [34]. Sediment sections for the pore water analysis were extruded into 50-mL centrifuge tubes in a N_2_-filled glove bag to minimize the iron oxidation and consequent P adsorption. After centrifugation for 10 min at 1000× *g*, the supernatants were syringe-filtered (pore size 0.45 µm).

### 2.4. Chemical and Biological Analysis

Concentrations of NH_4_^+^ and SRP were determined by colorimetric methods [35]. Both NO_3_^−^ and NO_2_^−^ were analyzed by ion chromatography [36]. Dissolved N_2_ and O_2_ concentrations were measured using the ratios of N_2_ to Argon (Ar) and O_2_:Ar using membrane inlet mass spectrometry, where the Ar concentration was assumed saturated [34]. DIC was quantified with an IR-based DIC analyzer (Apollo SciTech, Inc. ModelAS-C3, Newark, DE, USA). Flux rates of the solutes and gas were estimated as: (1)$$\mathrm{Flux}\;\mathrm{rate}=\frac{V}{S}\times \frac{dC}{dt}$$
where *C* is the pore water nutrient concentrations, *t* is the incubation time, *V* is the volume of the overlying water, and *S* is the area of sediment in the incubation cores.

The sediment water content was determined as the weight loss after drying at 65 °C for a week, and the grain size was determined by a sieving and pipette analysis [37]. The percent of water and the dry sediment density (r ~ 2.5 g cm^−3^) were used to calculate the porosity (∅) [38]:(2)$$\emptyset =\frac{water\%}{\left(water\%+\frac{\left(1-water\%\right)}{\rho }\right)}$$

Chl *a* was measured by a fluorometric analysis after acetone extraction [35]. The phytoplankton samples were examined microscopically. For phytoplankton identification and enumeration, Lugol’s iodine preserved samples were gently inverted, and a three-milliliter aliquot was settled for at least one hour in a Lab-Tek Chamber Slide^TM^ (Nalge Nunc #155379, Rochester, NY, USA). For the phytoplankton community analysis, nano- and microplankton (>3–200 μm) in the chamber were identified to the lowest taxonomic group possible using a Zeiss Axiovert 200 inverted microscope. Small and numerous phytoplankton species (at least 200 cells/sample, examined in a recorded number of optical fields) were identified and enumerated at 400× magnification; at 320×, a transect of the chamber was examined to capture the identification and enumeration data for moderately sized species, and the large/rare species in the entire chamber were identified and enumerated at 100× magnification.

An ambient water sample was also taken during the experiments for determination of the total microcystins at the Maryland Department of Natural Resources (MD DNR) Laboratory using the Adda Enzyme-Linked Immunosorbent Assay following EPA Method 546.

### 2.5. Statistical Analysis

The effects of Phoslock^®^ on P removal were assessed using a two-way ANOVA for the sampling time and treatment effects for SRP changes. Flux rates among the Phoslock^®^-free and 1st day and 28th day of Phoslock^®^ applications in the sediments were compared with the least significant difference at *p* < 0.05. All data analyses were conducted with SigmaPlot 14.0.

## 3. Results

### 3.1. Bloom Progression in the Lake

Cyanobacteria bloomed in Higgins Mill Pond from June to September. The cyanobacteria community was dominated by non-N_2_-fixing cyanobacteria (e.g., *Microcystis* spp.) in June and various N_2_-fixing cyanobacteria (e.g., *Anabaena* spp., *Anabaenopsis* spp., *Lyngbya* spp., *Chroococcus* sp., and *Pseudanabaena* sp.) from July to September. The Chl a concentrations exceeded 50 µg L^−1^. In the water column experiments, the initial cyanobacteria density and the bloom-driven changes were similar to those during the sediment capping experiments. In the closed-bottom experiment (no sediment nutrient supply), the initial total cyanobacteria density was 3.89 × 10^6^ cells mL^−1^, in which 56% of the cells were potential N_2_-fixing species. At the beginning of the open-bottom experiment (sediment nutrients released into the overlying water), the cyanobacteria density was 4.3 × 10^6^ cells mL^−1^, with 59% of the cells potential N_2_-fixing species. During both experimental periods in the lake, high pH (>9) and DO (>100 µmol L^−1^) were observed.

The microcystin levels (18 µg L^−1^, Table 1) exceeded the World Health Organization (WHO) 1998 chronic drinking water guidance value of 1 µg L^−1^ and were greater than the recreational safety guidance for children of 10 µg L^−1^. Table 1 provides the conditions in the treated and untreated pond samples and sediment compositions.

Due to the increased biological nutrient uptake, the inorganic nutrients in the lake decreased with the bloom development. NO_3_^−^ decreased from 34.6 µmol L^−1^ in June to being undetectable in August, the SRP reduced from 6.5 µmol L^−1^ to 1.1 µmol L^−1^, and the NH_4_^+^ decreased from 0.5 µmol L^−1^ to < 0.01 µmol L^−1^. As a result, the concentration ratios of DIN:SRP in our study period ranged from 0.23 to 10, reflecting a potential N limitation and consistent with the succession of diazotrophic cyanobacteria dominance over time.

### 3.2. Lake Water Treated with Phoslock^®^

Before the experiments, the concentrations of SRP and Chl *a* were 5.4 µmol L^−1^ and 50 µg L^−1^, respectively (Table 1). In the lake water, both SRP and Chl *a* were sustained around the initial levels, with the pH rising from 9.5 to 10.2 and DO supersaturated (120–250%) in the days after the initiation of the experiment.

In the closed-bottom mesocosms, the SRP, Chl *a*, and pH decreased in all the mesocosms and remained low in the Phoslock^®^-treated water without additional nutrient input (Figure 2 and Figure 3). One day after the Phoslock^®^ addition, the SRP concentrations decreased to 1.26 µmol L^−1^, 62.3% lower than the controls and 92% lower than the lake water (Figure 2). Over the next few days, the Chl *a* concentrations in the Phoslock^®^-treated mesocosms were 26–53% lower than the controls and 90–92% lower than the concentrations in the lake. In the Phoslock^®^-treated water, the pH decreased significantly from 9.5 to 7.4, while the pH increased from 8.9 to 9.9 in the nontreated control and from 9.42 to 9.97 in the lake water (Figure 3).

### 3.3. Sediment Pore Water Changes with Phoslock^®^ Treatment

The pore water SRP concentrations increased to >150 μmol L^−1^ with the depth, with the steepest gradients in the top 2 cm. The impact of Phoslock^®^ was dramatic at the sediment surface one day after Phoslock^®^ application, with the pore water SRP at near-surface horizons reduced to near the detection limit (0.002 µmol L^−1^). The SRP gradient below this thin Phoslock^®^ richlayer was barely affected (Figure 4A). In the long-term treatment, the pore water SRP concentrations in the Phoslock^®^-treated sediments exhibited the smallest concentration gradients (indicating a shift from soluble P to Phoslock^®^-bound P) within the 0~3-cm depth horizon. At 3 cm deep, the pore water SRP concentrations after a 28-day treatment were 72 µmol L^−1^, which was ~50% lower than the untreated samples and samples treated for a single day (150–156 µmol L^−1^, Figure 4A).

Although the pore water NH_4_^+^ (Figure 4B) increased gradually with the depth in all the sediment cores, the sediment capped by Phoslock^®^ for 28 days exhibited the lowest concentration gradients down to 3 cm deep, with the concentrations remaining lower than the control and 1-d Phoslock^®^ treatment. The concentrations of NO_3_^−^ (Figure 4C) within all the sediment cores showed a sharp decrease from ~1.2 µmol L^−1^ in the overlying water to nearly undetectable at ~1 cm deep in the sediment, indicating a downward flux from the overlying water into the sediment. The porosity (Figure 4D) decreased from 0.7 to 0.4 from the sediment surface to ~2 cm deep.

### 3.4. The Short-Term Effect of Phoslock^®^ on Sediment Flux

The changes in the nutrient fluxes (Figure 5A) were consistent with the vertical profiles in the pore water (Figure 4A). In the short term, Phoslock^®^ capping changed the N and P release from the sediments (Table 2). The Phoslock^®^-treated sediment cores were typified by a substantial P uptake (−15 µmol m^−2^ h^−1^) into the sediment (Figure 5A) in contrast to the P release (7.5 µmol m^−2^ h^−1^) in the untreated sediment.

In the control sediments, the fluxes of oxygen into the sediment averaged 911 µmol m^−2^ h^−1^ (Figure 5E, similar to the rates of DIC efflux (Figure 5F). Modest increases in the uptake of oxygen and efflux of DIC after 1 day of Phoslock^®^ addition were not significant. After 28 days, the rates of oxygen uptake and DIC efflux decreased significantly; these data suggest a decreased loading of organic material leading to lower rates of sediment metabolism. An inverse relationship between the oxygen consumption rates and DIC effluxes (linear regression coefficient (k) of −1.21, *p* < 0.01, Figure 5E,F) was observed.

The fluxes of NH_4_^+^ (82 µmol m^−2^ h^−1^) and N_2_-N (75 µmol m^−2^ h^−1^) were the major sediment–water exchange terms in the N cycle (Figure 5B–D). Both suggested that overlying water NO_3_^−^ was likely not the dominant source of N_2_-N production, and the coupled nitrification/denitrification was the dominant process. In terms of the N_2_-N loss to the total N fluxes, the denitrification efficiency accounted for 64.7%. While the uptake of NO_3_^−^ did not change in the short-term and longer-term observations (Figure 5C), the fluxes of N_2_-N decreased significantly with the addition of Phoslock^®^ to the core surface. Short-term Phoslock^®^ additions increased the ammonium efflux, though not significantly; the average N_2_-N decrease (48 µmol m^−2^ h^−1^) was similar to the average NH_4_^+^ increase (58 µmol m^−2^ h^−1^). Since there was little change in the influx of NO_3_^−^, these data suggest that short-term additions of Phoslock^®^ suppressed nitrification.

### 3.5. Long-Term Changes in Sediment Fluxes through Phoslock^®^ Capping

We compared the nutrient fluxes of sediments collected from the controls and the Phoslock^®^-treated open-bottomed limnocorral after 4 weeks (Figure 5). This comparison allowed us to assess whether the capping capacity was effective over longer time periods. The flux rates of SRP were 45% lower in the Phoslock^®^-treated sediments than in the controls (Figure 5A). Although the average NH_4_^+^ flux rates in the treated mesocosms were lower than the control rates after 28 days, the rates were not significantly different (Figure 5B): a slightly elevated NH_4_^+^ release after 1 d of Phoslock^®^ treatment and a decline in the rates similar to the control rates by 28 days. The Phoslock^®^ treatments did not significantly affect the benthic NO_3_^−^ fluxes (Figure 5C). The denitrification rates were higher after the 28-day Phoslock^®^ treatment than in the untreated controls or after 1 day in the treated samples (Figure 5D). The denitrification rates increased 44.5% relative to the observations in the untreated sediments and ~1.5-fold higher than the one-day capping. The denitrification efficiency at 28 days increased to 82.7% of the remineralized N. Relative to the control, the oxygen consumption rates and DIC flux rates decreased by 721-µmol O_2_ m^−2^ h^−1^ and 670-µmol C m^−2^ h^−1^, respectively (Figure 5E,F). Both illustrated a significant decrease in organic matter remineralization.

### 3.6. Water Column Changes after Phoslock^®^ Treatment on Sediments

Without the Phoslock^®^ treatments, cyanobacteria bloomed throughout the entire experiment period. The water column Chl *a* doubled, and the photosynthesis led to an elevated pH (9.5–10.2) and DO (400–580 µmol L^−1^, Figure 6). In the following days (days 10–50), the Chl *a* concentration (~24 µg L^−1^) in the treated samples remained 50% of the control’s (57 µg L^−1^).

The Phoslock^®^ treatment and the lower P availability were accompanied by a lower cyanobacteria abundance and altered speciation (Figure 7). In the treated water, the total cell numbers declined from the initial abundance of 6.1 × 10^6^ cell mL^−1^ to 1.3 × 10^5^ cell mL^−1^ in late August (50 days after sediment treatment). During this time, the abundance of N_2_-fixing cyanobacteria (mainly *Pseudanabaena* sp., *Lyngbya* sp., and *Chroococcus* sp.) decreased from 220 × 10^6^ cell mL^−1^ to 6.6 × 10^4^ cell mL^−1^. The abundance of *Microcystis* spp., the dominant non-N_2_-fixing species, was 2.8 × 10^3^ cell mL^−1^. These abundances were 23.5% and 6.2% of the numbers observed in the control water: 2.8 × 10^5^ cells mL^−1^ and 4.5 × 10^5^ cells mL^−1^ for the N_2_ fixers and non-N_2_-fixing cyanobacteria (e.g., *Microcystis* spp.), respectively.

## 4. Discussion

High nutrient loading, coupled with weak circulation and outflow and elevated summer temperatures, enhances the sedimentation and regeneration of N and P, along with their availability for phytoplankton uptake and subsequent growth in the lake. High P loading, coupled with nutrient sediment release, are the likely causal factors for cyanobacteria blooms in this shallow (<2 m) water. Based on the temperature profiles, the water was well-mixed. In response to N limitation throughout the summer (Table 2, N:P = 0.66), N_2_-fixing cyanobacteria (e.g., *Anabaena* spp.) dominated the lake ecosystem (unpublished data, MD DNR). Additionally present was a small fraction of non-N_2_-fixing cyanobacteria (e.g., *Microcystis* spp.).

### 4.1. Phoslock^®^ Influences on Water Chemistry

The results of this study are consistent with the previous results from Phoslock^®^ applications to wastewaters and eutrophic bloom waters [25,39]. The phosphate adsorption kinetics of Phoslock^®^ suggests a rapid equilibrium (<60 min) and high removal efficiency, with faster equilibrium rates and higher adsorption capacity at higher temperatures [40]. Simultaneous algal uptake may account for part of the observed decrease in SRP. Nonetheless, 20–30% higher P removal in the treated waters than in the controls suggests strong Phoslock^®^ adsorption. Elevated pH (>9) and DO (>100 µmol L^−1^) in the control groups resulted from intense photosynthetic carbon uptake and oxygen release. In contrast, the reduced P availability after the Phoslock^®^ treatments resulted in reduced Chl *a* concentrations and photosynthesis, leading to a decrease in pH and DO.

Regardless of the other natural effects (e.g., phytoplankton P uptake), we assume all the decrease in the P concentrations in water is due to adsorption. We calculated the maximum phosphorus absorbed on Phoslock^®^ from the reduction in P and the amount of absorbent. This value (~1.5 µmol-Pg^−1^) was far below saturation compared to the maximum adsorption capacities of ~100–140 µmol-Pg^−1^ [25], 255–330 µmol-Pg^−1^ [40], and 645 µmol-Pg^−1^ [29]. A large pool of ‘active’ P-bonding sites may have remained on the added Phoslock^®^ after sedimentation.

### 4.2. Changes in Sediment Fluxes One Day after Phoslock^®^ Addition

Once set on the sediment surface, Phoslock^®^ acted as a capping reagent to cover the lake sediment, blocking SRP release from the sediment into the water column. The thickness of the capping layer was ~1.5 mm, similar to the depth noted by Vopel et al. (2008) [41] for the application of 135–700-g m^−2^ Phoslock^®^. Their microelectrode measurements revealed that Phoslock^®^ hindered DO penetration into the sediment and raised the location of the oxic–anoxic interface from several millimeters below the natural sediments into the capping layer. This was associated with a decrease in pH, ~0.5–1 units lower at the initial sediment surface (now below the sedimented Phoslock^®^ layer) than the topmost Phoslock^®^ layer [41]. These changes, in turn, exerted temporal influences on the P release and N cycling in the sediments.

Normally, phosphorus from organic matter degradation in sediments is released into pore water [42]. It then diffuses upward into the oxic sediment surface, where it can be sequestered by iron oxides. However, Phoslock^®^ has a much stronger adsorption capacity than the natural oxic layer of the sediments, leading to a reduction in SRP fluxes across the sediment–water interface [43,44]. Moreover, pH decreases also favor the Phoslock^®^-binding efficiency.

As the pH decreased from 9.4 to 7.3 as the phytoplankton reduced after treatment, the main phosphate species changed from HPO_4_^2−^ to H_2_PO_4_^−^. As well, the lanthanum in Phoslock^®^ was converted from insoluble La(OH)_3_ to La(OH) _2_^+^, which can absorb more phosphorus [40]. Vertically, the sediment pH decreased from >9 at the sediment–water interface to ~6 within the sediments during bloom progression [33,45]. This change would raise the P adsorption capacity by 30% [25,41]. The adsorption of SRP on Phoslock^®^, derived either from the bioavailable P in the water or P intercepted as it diffuses upward in the pore water, may cause a net P sequestration and sink in the sediments.

The redox environment is a key determinant in biogeochemical N cycling. A portion of the NH_4_^+^ regenerated from organic matter decomposition is oxidized to NO_3_^−^ (nitrification) in the upper few millimeters of the sediment before escaping from the sediment [42,46,47]. Denitrification occurs anaerobically below the oxic layer, where the NO_3_^−^ or NO_2_^−^ from nitrification serves as a terminal electron acceptor [48].

After the Phoslock^®^ addition, remineralized N was increasingly released as NH_4_^+^ rather than N_2_ via the nitrification–denitrification process. The condensed Phoslock^®^ slightly slowed the oxygen penetration from the overlying water; thus, a thinner oxic layer may further reduce the possibility for ammonium oxidation (nitrification) and may explain the enhanced NH_4_^+^ flux rates. Moreover, one day maybe not be sufficient for denitrifying bacteria to populate near the new oxic–anoxic boundary in the sediment, especially within the organic matter-poor Phoslock^®^ layer [49].

### 4.3. Long-Term Changes in Sediment Fluxes after Adding Phoslock^®^

The long-term changes of the SRP pore water profiles and flux rates were consistent with previous studies. Meis et al. [50] reported that La, the main P-binding element in Phoslock^®^, can be vertically transported by bioturbation or wind-induced sediment resuspension and, thus, disperse La into the top 0–4 cm or even deeper into the sediments one month after Phoslock^®^ is applied. In this study, after Phoslock^®^ was introduced, and the observed concentrations of the pore water SRP were reduced within the top 2 cm sediments where the porosities were larger than the deeper sediments, reflecting either the downward movement of Phoslock^®^ over time or the depuration of diffusive P via enhanced adsorption. This may lead to a decrease of the pore water phosphate concentration at the top of the sediment column by absorbing the labile P (e.g., from pore water and loosely bound P on sediment particles) and forming a refractory P fraction [50]. Moreover, the decreased pH from >9.5 to ~8 in the water after long-term treatment, coupled with the further decrease in the sediment to pH < 7 by adding a Phoslock^®^ layer, led to a better affinity of Phoslock^®^ to monovalent dihydrogen phosphate and a reduction in the SRP flux rates [25,40].

In response to long-term capping, diminished phytodetritus sedimentation and pH may contribute to the reduced NH_4_^+^ flux and enhance the N_2_ release. In the Phoslock^®^-treated limnocorral, the diminished biomass may reduce the accumulation of phytodetritus on the sediment surface. The decreased rates of oxygen uptake and DIC efflux are consistent with this. Conversely, in the controls, the enriched organic matter may result in faster nutrient fluxes and more rapid rates of respiration [51,52].

In lakes, the penetration of high pH from the overlying water into sediments may decrease P adsorption to Fe-oxides and enhance the total ammonium (NH_4_^+^ and NH_3)_ diffusion gradients, elevating fluxes of both P and N [19]). A high pH (9–9.5) can inhibit nitrifying bacteria activity [53], nitrification efficiency [54], and N_2_ loss through denitrification [33,55]. Moreover, bentonite contributes ~95% of the material in Phoslock^®^ [43]. It may reduce the NH_4_^+^ release via adsorption and impact the downstream nitrogen fluxes. Bentonite particles display a negative charge at a wide range of ion strength and pH (4–9) [56] and are used to absorb NH_4_^+^ in soil and groundwater [57]. Absorbed NH_4_^+^ is theoretically considered as a substrate for nitrification [58], and the anoxic habitats after administration may favor denitrifying bacteria [59]. The remediation-related changes, together with the weak pH penetration, likely enhance the N_2_ loss from ecosystems.

### 4.4. Ecological Response due to Phoslock^®^

In the long-term incubation, the positive feedback was built up after the Phoslock^®^ application for the first ~20 days. With the N_2_-fixing capacity in N-limited conditions, diazotrophs have a relatively stronger demand for P than N from ambient water. After remediation, P removal from water tended to reduce the bioavailable P, which limited the cyanobacteria growth. The pH fell from abnormally high (>9) to ~8, diminishing the release of sedimentary phosphate driven by high pH [19]. Although no direct measurements were made in this study, diazotrophic cyanobacteria have been shown to maintain the ambient phytoplankton communities by releasing fixed N [16,60]. The loss of N_2_-fixers, combined with a limited DIN release from the sediments, may finally cause bloom cessation.

Based on the SRP flux measurements in the field (~10 µmol m^−2^ h^−1^) and the P-binding capacity from 20-g P/kg Phoslock^®^ [29] to 50-g P/kg Phoslock^®^ [61], the amount applied in this experiment would have reached saturation in 20–50 days if the P capping was completely efficient and the external inputs were reduced. Comprehensive assessments of both the pH and oxic–anoxic effects on nutrient diffusion and P adsorption–desorption in the sediments should be further conducted to optimize the amount and frequency of Phoslock^®^ treatment to manage the environmental effects of cyanobacteria blooms.

## 5. Conclusions

In this study, the addition of Phoslock^®^ lowered the dissolved P concentration and mitigated the CyanoHABs. Phoslock^®^ rapidly and efficiently adsorbed SRP and reduced the cyanobacterial populations. Phoslock^®^-capped sediments served as a barrier to slow P diffusion from the sediments into the overlying waters. At the same time, the postulated upward movement of the oxic–anoxic interface through the sediment column slightly enhanced the NH_4_^+^ release and temporarily inhibited the cycle of nitrification and denitrification, with denitrification reaching the highest rates 4 weeks after treatment. Phoslock^®^ additions, together with reduced pH and organic matter sedimentation, created a positive feedback loop to reduce the internal nutrient input from the sediment into the water and promoted nitrogen loss associated with coupled nitrification and denitrification.

CyanoHABs pose huge problems globally, now exacerbated by increasing water temperatures, population increases, and food production that deliver increasing nutrients to fresh and marine waters. Phoslock^®^ is one technology that prevent blooms, as its role is to bind soluble P and limit blooms. It works well when the external P supplies are eliminated but can saturate quickly if the watershed loads of P remain high. Hence, external loads determine Phoslock^®^’s efficacy, with repeated and costly applications if the external P supplies remain substantial. However, reducing the P (and N) loads in watersheds is extremely costly, as land use changes must occur. That requirement curtails the effective prevention of blooms, as modifying sufficient land areas to reduce nutrient supplies remains beyond most community’s fiscal resources, political will, and land access. For internal P, dredging is also costly and suffers from the same problem, i.e., can external P loads be curtailed? The direct intervention of CyanoHABs is often pursued, with short-term success for small-to-medium-sized ponds, lakes, and reservoirs. Some interventions can persist for years (e.g., granular peroxide control of *Planktothrix agardhii*, Lake Anita Louise, MD, USA [62]); others are effective at annual intervals (e.g., barley straw application for Microcystis in Lake Williston, MD, USA [63]), while others have short-to-long-term efficacies (e.g., clay flocculation and sediment capping [64]). The costs for these relative to Phoslock^®^ are clays > Phoslock^®^ > peroxide >> barley straw. For additional detail, see the reviews by Anderson et al. [65] and Sellner and Rensel [66], as well as a list and review of the many methods, strategies, and technologies at https://hcb-1.itrcweb.org/(accessed on 12 December 2021).

## Figures and Tables

**Figure 1 ijerph-18-13360-f001:**
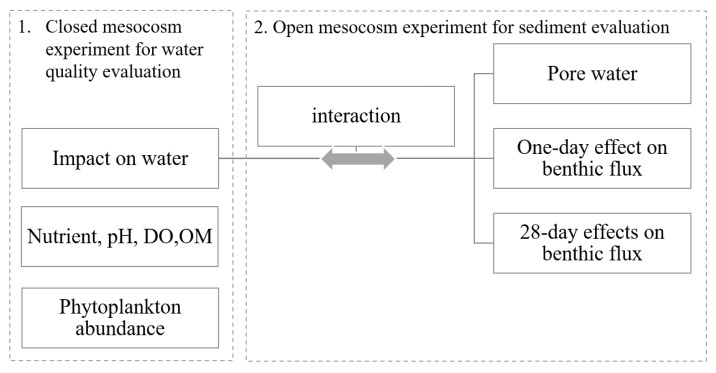
Experimental design.

**Figure 2 ijerph-18-13360-f002:**
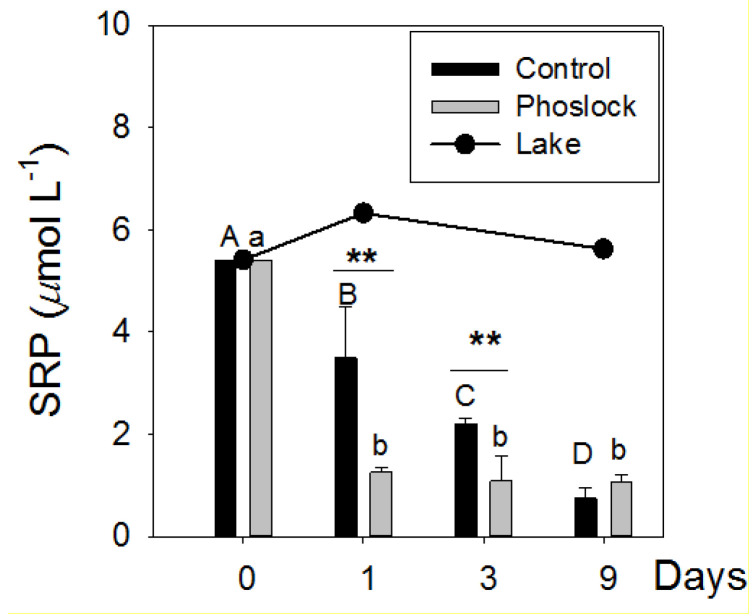
Changes of soluble reactive phosphorus (SRP ± standard error) after Phoslock^®^ treatment. Data collected from lake water (*n* = 1) and from closed-bottom mesocosms for the controls (*n* = 3) and the Phoslock treatments (*n* = 3). Capital letters and lowercase present the differences over time in the control and Phoslock-treated mesocosms (*p* < 0.05), respectively. ** The differences between treatments at the same sampling time (*p* < 0.01).

**Figure 3 ijerph-18-13360-f003:**
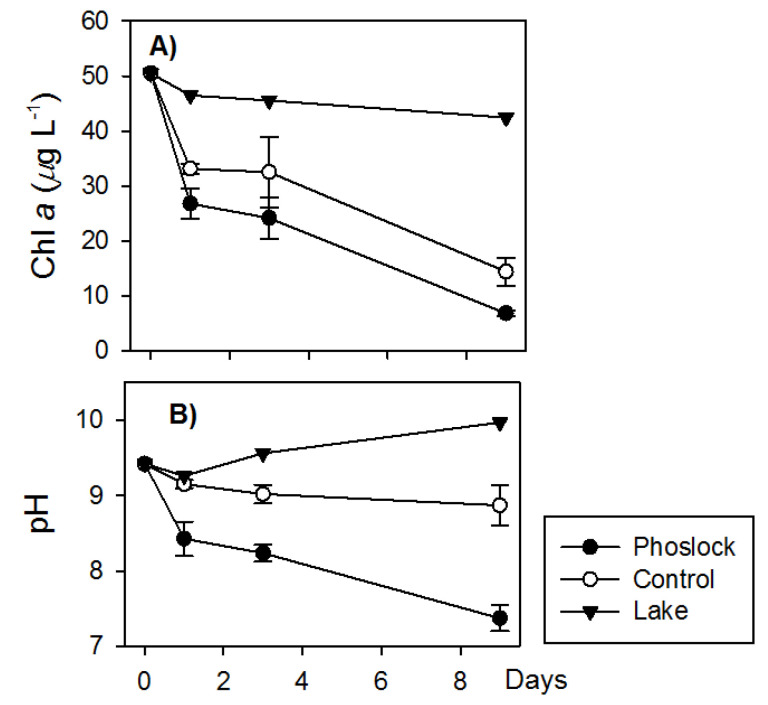
Changes in Chlorophyll *a* (**A**) and pH (**B**) over time in the lake water and the Phoslock^®^-treated and untreated mesocosms. Data were collected from the lake water (*n* = 1) and from the closed-bottom mesocosms for the controls (*n* = 3) and Phoslock^®^ treatments (*n* = 3) from 2 July to 11 July. Error bars indicate standard errors.

**Figure 4 ijerph-18-13360-f004:**
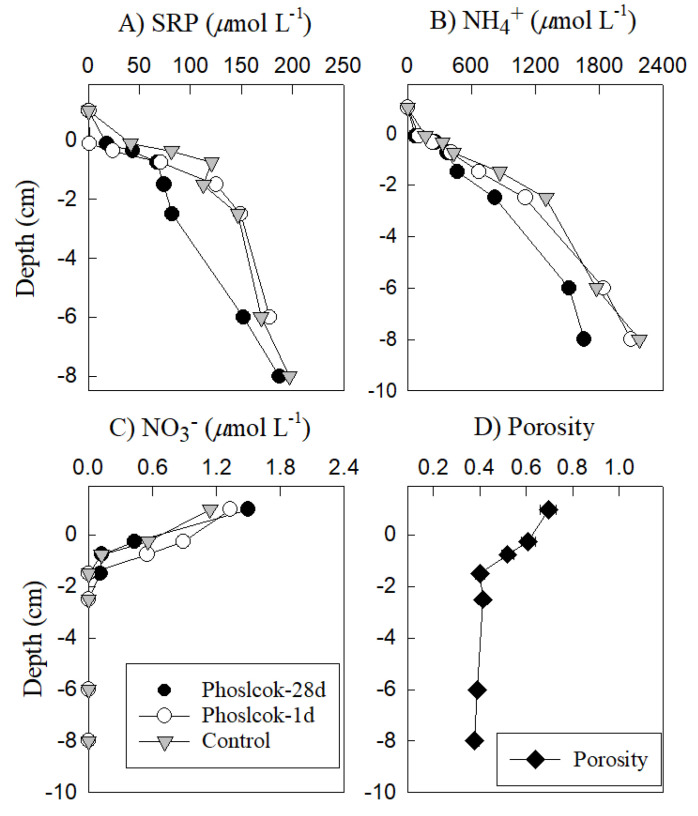
Vertical profiles of the pore water nutrients and porosity taken from Phoslock^®^-free sediments (control) and from sediments capped with Phoslock^®^ for 1 day and 28 days. The top symbol in each panel represents the conditions in the pond water immediately above the sediment. The pore water nutrients include SRP (**A**), NH_4_^+^ (**B**), NO_3_^−^ (**C**), and Porosity (**D**).

**Figure 5 ijerph-18-13360-f005:**
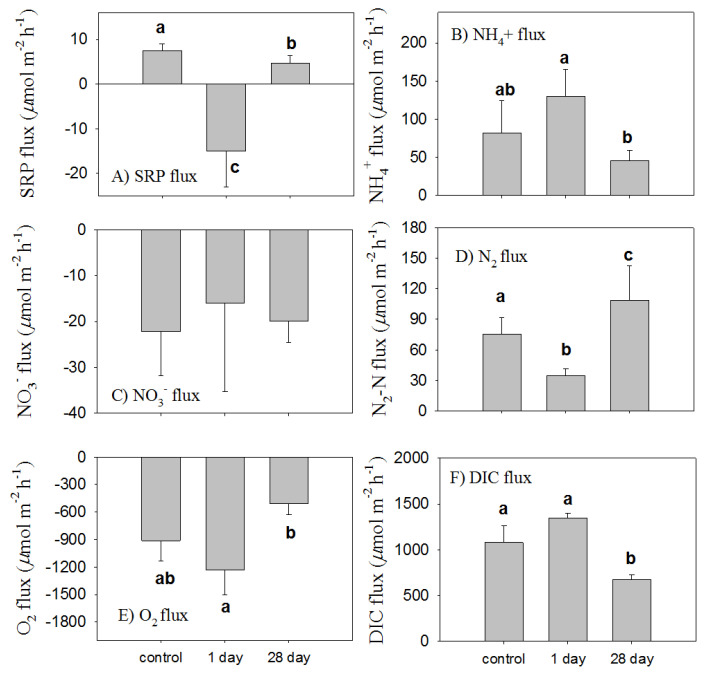
Flux rates at the sediment–water interface in Phoslock^®^-free and 1 day and 28 days after Phoslock^®^ treatment. Flux rates include inorganic nutrient flux (**A**–**C**), denitrification (**D**), sediment oxygen consumption (**E**), and respiration rates (**F**). Symbols a, b, and c present the differences between means at a significance level of *p* < 0.05.

**Figure 6 ijerph-18-13360-f006:**
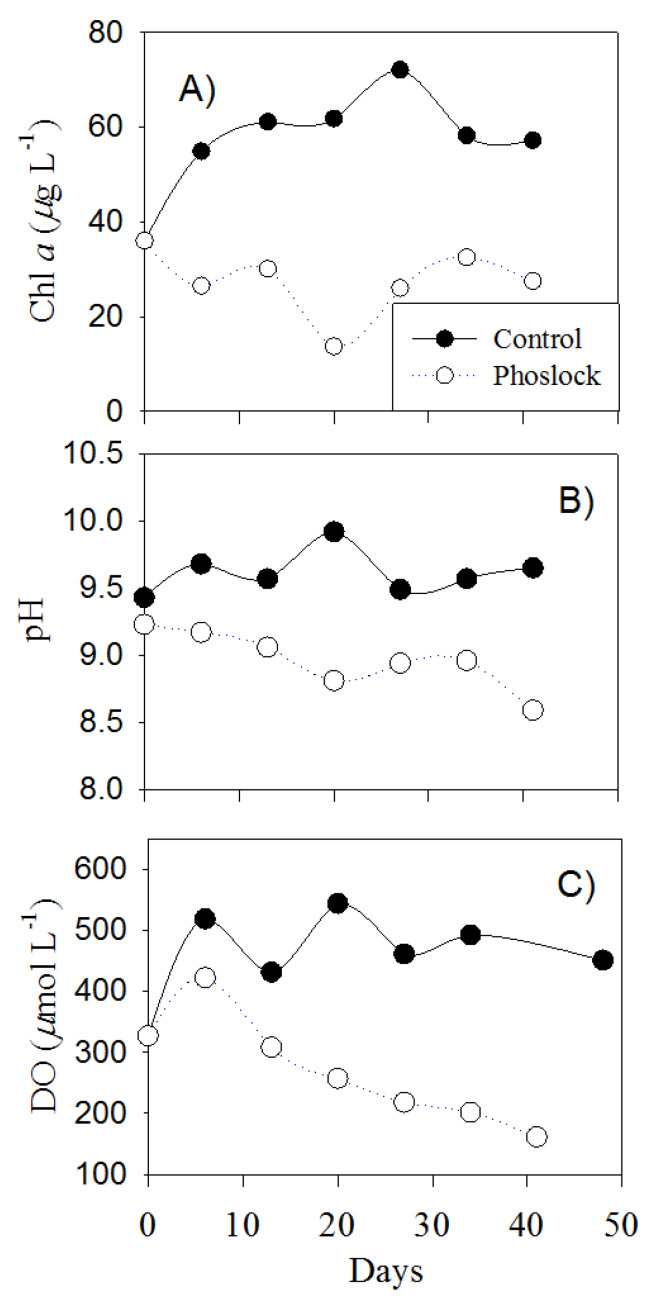
Water quality improvement after Phoslock^®^ treatment, including changes in Chlorophyll *a* (**A**), pH (**B**), and dissolved oxygen (**C**). Water samples were collected for ~50 days after the Phoslock^®^ addition.

**Figure 7 ijerph-18-13360-f007:**
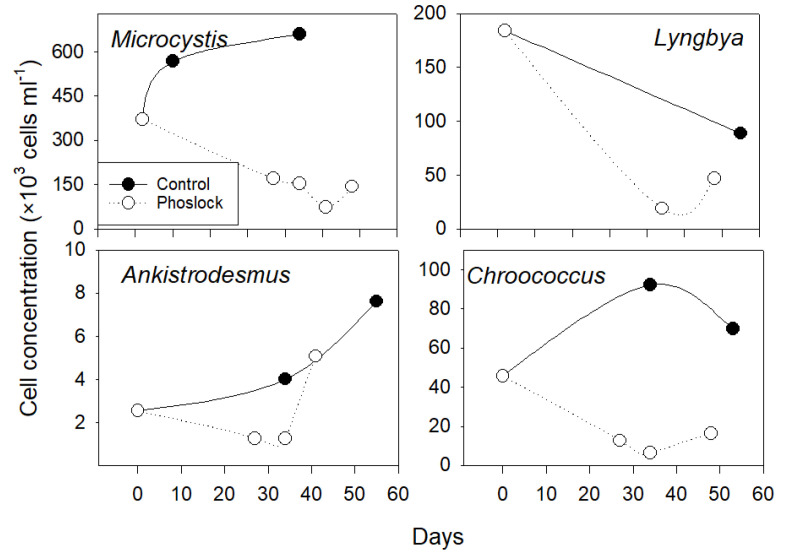
Cyanobacteria species changes over time in the lake and treated limnocorral. Data was collected from 25 July to 12 September.

**Table 1 ijerph-18-13360-t001:** Experiment details.

Experiment Information		Phoslock^®^ Application	
-	-	Water	Sediment
time		2–11 July	26 July–12 September
container		closed-bottom	open-bottom
sediment nutrient input		no	yes
Initial conditions in lake water (mean ± standard error)			
temperature (°C)		32.5 ± 2.77	27.6 ± 2.19
salinity		0.2 ± 0.01	0.2 ± 0.02
pH		9.6 ± 0.1	9.5 ± 0.13
microcystin (µg L^−1^)		18.0 (*n* = 1)	7.8 ± 2.83

**Table 2 ijerph-18-13360-t002:** Bottom water and sediment properties with/without the Phoslock^®^ treatment on 22 August at Higgins Mill Pond.

Parameters	Phoslock^®^	Controls
Properties in water column—		
temperature (°C)	25.79	26.05
pH	8.80	8.97
DO (µmol L^−1^)	97.81	149.06
Chl *a* (ug L^−1^)	32.70	69.80
SRP (µmol L^−1^)	1.47	1.13
NO_3_^−^ (µmol L^−1^)	-	0.48
NO_2_^−^ (µmol L^−1^)	0.22	0.22
NH_4_^+^ (µmol L^−1^)	0.06	0.05
Sediment—		
sand (%)	18.52 ± 0.31	
silt (%)	56.24 ± 0.36	
clay(%)	25.31 ± 0.43	

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
