# Peer review of "Mitigation of CyanoHABs Using Phoslock® to Reduce Water Column Phosphorus and Nutrient Release from Sediment"

_ijerph, 2021, doi:10.3390/ijerph182413360_

Round 1

Reviewer 1 Report

This manuscript by author provides an interesting subject about Mitigation of CyanoHABs Using Phoslock® to Reduce Water Column Phosphorus and Nutrient Release from Sediment

, and I can say the same things about the manuscript.

Detailed comments:

1. What is innovation of this M.S., author should give the more innovation clearly in introduction part? And there are many similar researches, what is the news?

2. In short-term (1 day) incubation experiments, Phoslock® was capped with sediments diminished or reversed SRP effluxes. At the same time, the upward movement of the oxic-anoxic interface through the sediment column slightly enhanced NH4+ release and inhibited the cycle of nitrification and de-nitrification. About this aspect, author should give more information and data.

3. In a long-term (28 days) experiment, Phoslock® hindered P release, reduced cyanobac- terial abundance, and alleviated the bloom-driven enhancements in pH and oxygen. Phoslock® ad-ditions, together with reduced pH and organic matter sedimentation, what is its scientific mechanism?

4. Phoslock® created a positive feedback loop to reduce internal nutrient input from sediment into the water and promoted nitrogen loss associated with the coupling nitrification-denitrification, Is this a new results?

Author Response

We thank the reviewer for their careful reading of the manuscript and their constructive remarks. We have taken the comments on board to improve and clarify the manuscript. Please see attached PDF file for a detailed response to all comments:

Reviewer 2 Report

The aim of the manuscript is mitigation of CyanoHABs using Phoslock® to reduce water column phosphorus and nutrient release from sediment in a mesocosm study.

The topic of this manuscript is interesting, and it contains a great amount of interesting data. However, some points deserve to be a little more discussed by the authors in order to improve the quality of the manuscript before publication. In this study, the authors focused only on cyanobacteria and no information on the impact of the Phoslock® on other non-toxic phytoplankton species. For example, the use of copper sulphate is very effective in controlling the proliferation of cyanobacteria, but it is not selective and also inhibits other non-toxic algae, which are necessary for biodiversity in an aquatic ecosystem. Other techniques have also been reported in the literature such as biological control by the introduction of planktivorous fish, the use of extracts of algae and aquatic plants, etc.I would appreciate that in discussion section, the authors do not focus the discussion only on the use of Phoslock®, but compare its advantages and disadvantages to other means of controlling cyanobacteria widely described in the literature.

Another point deserves to be clarified more concerning the quantification of phytoplankton. Lines 102-103, authors reported that water samples were also fixed with acid Lugol’s solution to quantify phytoplankton abundance. A few lines later (109-111), acid Lugol’s-preserved phytoplankton samples were processed weekly to identify cyanobacteria species and estimate cell abundance. So, in this study only cyanobacteria species were identified and quantified and not all phytoplankton. In addition, in the result section, the authors estimated the abundance of cyanobacterial species in Cell/L. Since they are colony species like Microcystis and filamentous species like Lyngbia, how the authors counted cells? Especially for filamentous species which is not so easy from a practical point of view. For filamentous species, Cell/L means cells or filament. It is for this reason and in recent years, the abundance of cyanobacteria has often been estimated in biovolume or biomass and is no longer in cell / L. It should be very interesting with the chlorophyll value, which indicates all the phytoplankton to also estimate phycocyanin, which is a biomarker of cyanobacteria. The ratio between the two pigments could give an idea on the impact of the use of Phoslock® on the development of cyanobacteria compared to the rest of the phytoplankton.

Author Response

We thank the reviewer for their careful reading of the manuscript and their constructive remarks. We have taken the comments on board to improve and clarify the manuscript. Please see attached PDF file for detailed response to all comments:

Round 2

Reviewer 1 Report

After revision, the quality of the article has been greatly improved. It will be published after further highlighting the innovation.

Author Response

Dear Reviewer,

We thank all your comments which have led to the great improvement of this draft. We added the flowing information to address the innovation:" In the previous studies, the assessment of Phoslock® application has focused on its performance (direct P adsorption, P removal efficiency in wastewater treatment) [25,27], the equilibrium and kinetics strength of La-P in response to water column changes (e.g. temperatures, ionic strength, and pHs)[25], and the reduced P release after capping on sediment in the field [31]. There is a general consensus that prevention is the preferred management strategy for HABs, but this management strategy is usually  difficult to implement. The whole ecosystem responds after application of Phoslock®remains questionable. In order to provide a sustainable solution for bloom control, we consider both N and P internal cycles, the induced N:P stoichiometry changes, and the mechanisms regulating phytoplankton biomass and speciation. The long term impacts on the ecosystem after Phoslock® application was also assessed."

Reviewer 2 Report

I thank the authors for their responses to the comments raised by the reviewers. These have improved the quality of the manuscript which is now acceptable for publication. 

Author Response

We thank the reviewer for the thoughtful comments which led to the great improvement of this manuscript.